# The Anti-Mullerian Hormone as a Biomarker of Effectiveness of Metformin Hydrochloride Therapy in Polycystic Ovarian Syndrome and Insulin Resistance

**DOI:** 10.3390/healthcare13080884

**Published:** 2025-04-11

**Authors:** Nikoleta Parahuleva, Anna Mihaylova, Stanislava Harizanova, Yana Merdzhanova, Mariya Koleva, Vasil Madzharov, Gergana Strikova, Ekaterina Uchikova

**Affiliations:** 1Department of Obstetrics and Gynecology, Medical Faculty, Medical University of Plovdiv, 4002 Plovdiv, Bulgaria; nikoleta.parahuleva@mu-plovdiv.bg (N.P.); yana.merdzhanova@mu-plovdiv.bg (Y.M.); ekaterina.uchikova@mu-plovdiv.bg (E.U.); 2Department of Health Care Management, Faculty of Public Health, Medical University of Plovdiv, 4002 Plovdiv, Bulgaria; 3Department of Hygiene, Faculty of Public Health, Medical University of Plovdiv, 4002 Plovdiv, Bulgaria; stanislava.harizanova@mu-plovdiv.bg; 4Department of General and Clinical Pathology, Medical Faculty, Medical University of Plovdiv, 4002 Plovdiv, Bulgaria; mariya.koleva@mu-plovdiv.bg; 5Department of Organization and Economics of Pharmacy, Faculty of Pharmacy, Medical University of Plovdiv, 4002 Plovdiv, Bulgaria; vasil.madzharov@mu-plovdiv.bg; 6Medical College, Medical University of Plovdiv, 4002 Plovdiv, Bulgaria; gergana.strikova@mu-plovdiv.bg

**Keywords:** polycystic ovarian syndrome, anti-Mullerian hormone, insulin resistance, metformin treatment, biomarker, metabolic syndrome

## Abstract

**Background/Objectives**: Among the therapeutic options available for managing PCOS, metformin improves insulin sensitivity, reduces androgen levels, and helps restore menstrual regularity and ovulation. While primarily used for its metabolic effects, metformin therapy may also influence reproductive parameters, including AMH levels, which are pivotal in improving ovarian function and predicting therapeutic outcomes in PCOS. The aim of this study was to search the scientific literature and analyze the correlation between AMH levels and metformin hydrochloride therapy in women with PCOS and IR. **Methods**: A systematic review of the scientific literature was conducted using the following keywords: polycystic ovarian syndrome, anti-Mullerian hormone, insulin resistance, metformin, treatment, biomarker, and metabolic syndrome. This review was aimed at investigating the potential of AMH as a biomarker of the effectiveness of metformin therapy in patients with PCOS and IR. **Results**: Metformin treatment in PCOS patients has shown significant reductions in serum AMH levels with prolonged therapy. As an insulin sensitizer, metformin improves insulin sensitivity, reduces hyperinsulinemia, and suppresses hyperandrogenism. This process inhibits the growth of antral follicles, which is reflected in decreased AMH levels. **Conclusions**: Reductions in AMH levels and improvements in insulin sensitivity can serve as indicators of treatment efficacy and enhancements in reproductive function for these patients. AMH could be considered a prognostic marker for evaluating the effectiveness of metformin therapy. A decrease in AMH levels following treatment may indicate improved ovarian function and a reduction in polycystic morphology. However, further research is necessary to confirm these findings and to determine the optimal dosages and duration of treatment.

## 1. Introduction

Polycystic ovary syndrome (PCOS) is one of the most common endocrine disorders affecting women of reproductive age, with approximately 8–20% of women in this group being affected, according to the diagnostic criteria [1]. Diagnostic criteria for PCOS are most often based on the Rotterdam criteria (2003), and at least two of the three criteria must be present for a diagnosis to be made: oligoovulation or anovulation—irregular or absent ovulation; hyperandrogenism; and polycystic ovaries on ultrasound—enlarged ovaries (≥10 cm^3^) and/or ≥12 small follicles (2–9 mm) in each ovary. The US National Institutes of Health (NIH, 1990) requires oligoanovulation and hyperandrogenism, but does not include the ultrasound criteria [2,3,4]. This condition is associated with hormonal and metabolic disturbances and an abnormal ovulatory cycle, often leading to infertility and various long-term health risks [5]. The pathophysiology of PCOS primarily revolves around hormonal imbalances, chronic low-grade inflammation, insulin resistance, and hyperandrogenism, which disrupt folliculogenesis and elevate the risk of associated comorbidities, including type 2 diabetes and endometrial cancer [6]. One of the key hormonal markers in the pathophysiology of PCOS is anti-Müllerian hormone (AMH), which plays a central role in the development of ovarian follicles and the regulation of ovulation [7,8,9,10,11,12,13]. AMH suppresses the recruitment of primordial follicles into the growing pool, thus ensuring that only a subset of follicles mature during each cycle. It inhibits the action of follicle-stimulating hormone (FSH) on early-stage follicles, thereby controlling the rate of follicular development. As follicles mature, AMH production decreases, and it is undetectable in fully mature follicles [14].

AMH is widely used to assess ovarian reserves. Its levels decline with age as the pool of viable follicles diminishes. In clinical practice, measuring AMH is a useful indicator of reproductive potential, particularly in assessing conditions like diminished ovarian reserve and PCOS [15]. High AMH levels often correlate with a larger number of antral follicles, while lower levels are associated with reduced fertility [16].

AMH secretion in females is influenced by several factors, including gonadotropins like FSH. Elevated FSH levels downregulate AMH secretion, whereas lower FSH levels allow for higher AMH production [14]. Age is another significant factor; as women age, their AMH levels naturally decrease due to the declining number of antral follicles [16].

Insulin resistance (IR) is a key mechanism associated with polycystic ovary syndrome, affecting approximately 35–80% of women with the condition [17]. Elevated insulin levels stimulate androgen production in the ovaries and reduce sex hormone-binding globulin (SHBG), leading to hyperandrogenism—a hallmark feature of PCOS [18].

Metformin is a widely used medication primarily prescribed for the management of type 2 diabetes, but it has also been studied for its potential effects on reproductive hormones, including anti-Müllerian hormone [19]. AMH plays a key role in ovarian function, acting as a marker of ovarian reserve and regulating follicular development. Research has shown that metformin may influence AMH levels, particularly in conditions such as polycystic ovary syndrome, a common cause of infertility, where AMH levels are typically elevated. Metformin hydrochloride is also a treatment used for insulin resistance in women with PCOS, and it has been shown to effectively reduce androgen levels and improve ovulatory function [20].

In women with PCOS, AMH levels are often higher than normal due to an increased number of small antral follicles in the ovaries. Metformin, by improving insulin sensitivity and reducing hyperinsulinemia, may have an indirect effect on AMH levels. Studies have shown that metformin treatment can lead to a reduction in AMH levels in women with PCOS, possibly by reducing the number of small, underdeveloped follicles in the ovaries [21]. This reduction in AMH levels could reflect an improvement in the regulation of folliculogenesis, leading to a more balanced ovarian environment.

Metformin’s mechanism of action is thought to involve several pathways. Primarily, metformin improves insulin sensitivity, which in turn reduces circulating insulin levels. High insulin levels are known to exacerbate ovarian dysfunction in PCOS by promoting the excessive production of androgens and affecting the ovarian follicular reserve. By decreasing insulin resistance, metformin may help normalize AMH levels by reducing the number of small antral follicles that are typically associated with elevated AMH [22].

Additionally, metformin may reduce the activity of the mTOR (mechanistic target of rapamycin) pathway, which is involved in cell growth and proliferation. In the context of the ovaries, mTOR is thought to play a role in the survival of early-stage follicles. By modulating mTOR, metformin could influence the recruitment and development of follicles, leading to a decrease in AMH levels as ovarian function normalizes [23].

Metformin treatment may reduce serum AMH levels in patients diagnosed with PCOS, taking into account the age of the patients and AMH levels, as well as the dose and duration of metformin treatment, according to a meta-analysis by Zhou et al. (2023) [24]. Other studies suggest that metformin treatment leads to a reduction in AMH levels, which may indicate improvement in ovarian function and hormonal balance normalization [25].

Concentrations of serum AMH levels are higher in women with PCOS [26,27], consistent with increased numbers of preantral and microantral follicles that produce AMH [28]. Elevated serum AMH in women with hyperandrogenism or oligocystic ovarian syndrome may indicate to a clinician the presence of PCOS in the absence of a reliable ultrasound [29].

Burghen et al. described the presence of hyperinsulinemia in polycystic ovarian syndrome (PCOS) and the positive correlation between hyperandrogenism and insulin levels [30]. Androgen synthesis insulin resistance is stimulated by the direct influence of hyperinsulinemia on androgen metabolism, which occurs due to stimulation of enzyme activity, as a result of which androstenedione synthesis is increased. Insulin has another mechanism which leads to hyperandrogenemia, i.e., inhibiting SHBG production in the liver. On the other hand, this leads to an increased concentration of free testosterone. It has been proved that under the conditions of hyperinsulinemia, sensitivity to insulin and glucose utilization is decreased only in peripheral tissues, for example, muscular tissue, whereas there is no decrease in insulin levels in the ovaries [20]. Several studies support the positive correlation of androgens containing AMH in serum [31,32] with over-production of androgens, including intrinsic defects in theca-cells, in PCOS [33].

Among the therapeutic options available for managing PCOS, metformin hydrochloride has emerged as a cornerstone of treatment, particularly for its role in managing insulin resistance—a common feature in PCOS patients. Metformin, an oral hypoglycemic agent, improves insulin sensitivity, reduces androgen levels, and helps in restoring menstrual regularity and ovulation. While primarily used for its metabolic effects, metformin therapy may also influence reproductive parameters, including AMH levels, which are pivotal in understanding ovarian function and predicting therapeutic outcomes in PCOS [34].

Metformin (1,1-dimethylbiguanide hydrochloride) is an oral antidiabetic medication within the biguanide group. The US FDA has affirmed it for treating T2 diabetes. It is typically prescribed at daily doses ranging from 500 to 2550 mg [12,35]. Its essential clinical activity is to restrain hepatic glucose generation, in spite of the fact that it diminishes intestinal glucose take-up, and increases in take-up negatively influence affectability in fringe tissues. Metformin has antilipolytic impacts, bringing down circulating free greasy corrosive concentrations, which eventually helps in diminishing gluconeogenesis [36]. Metformin plays a crucial role in PCOS by reducing insulin levels, which leads to decreased luteinizing hormone and androgen levels [12].

This study focuses on the effect of metformin hydrochloride on AMH levels in the management of PCOS. By examining changes in AMH as a response to metformin therapy, the research sought to elucidate potential mechanisms by which metformin improves ovarian function and overall reproductive health in women with PCOS. Understanding this connection could offer deeper insights into the role of AMH as a biomarker for evaluating treatment effectiveness and as a valuable tool for developing personalized therapeutic approaches in the management of PCOS.

The aim of this study was to search the scientific literature and analyze the correlation between AMH levels and metformin hydrochloride therapy in women with PCOS and IR. Research into this mechanism could provide new insights into using AMH as a biomarker of the effectiveness of metformin treatment and for personalizing therapeutic strategies in PCOS management.

## 2. Materials and Methods

The Preferred Reporting Items for Systematic Reviews and Meta-Analyses (PRISMA) guidelines were followed to perform this work [37].

### 2.1. Literature Search

The authors systematically and comprehensively compiled all relevant studies on the molecular classification of endometrial carcinoma and its precursor lesions within the medical and healthcare fields. Data were retrieved from the Web of Science, PubMed, Scopus, Google Scholar, and ScienceDirect databases. A strategy for structured search was employed using the following keywords: polycystic ovarian syndrome, anti-Mullerian hormone, insulin resistance, metformin, treatment, biomarker, metabolic syndrome, and, in addition, (opportunities) or (advantages) and (review) or (systematic review), to locate articles discussing the relationships between AMH, insulin resistance, and polycystic ovary syndrome.

### 2.2. Eligibility Criteria

The inclusion criteria encompassed studies published between 2000 and 2025, including reviews, systematic reviews, meta-analyses, and full-text articles. The inclusion criteria included articles with information on patients with medium and long-term metformin use (3–6 months and over 6 months), as well as average and maximum daily doses of administration (1500–2500 mg/day). The age of the women was not an exclusion criterion, and the data included information on patients of all age groups. Only clinical studies were included; preclinical studies were excluded.

### 2.3. Exclusion Criteria

Exclusion criteria comprised abstracts, short communications, patents, policy-related documents, case reports, and studies lacking essential information. Articles were included irrespective of language restrictions, and a comprehensive summary of the results was compiled. Duplicate records were removed, and records were excluded due to insufficient data or discrepancies in the study design. During the process, there were other criteria for exclusion:Case reports.Reports of preclinical studies.Reports of complications not directly related to the main objective of the study.

### 2.4. Data Analysis

We utilized Microsoft Office Excel 2010 to develop a standardized data extraction form, facilitating systematic retrieval and analysis of data. Articles were sourced from multiple databases and compiled in an Excel file, with duplicate entries removed. Subsequently, two independent authors reviewed the abstracts of the retrieved studies, selecting a subset for further evaluation. The full texts of these selected articles were then independently assessed, enabling a final determination of relevant studies. A limited number of experimental and prospective studies met all the predefined inclusion and exclusion criteria.

A total of 812 potentially relevant articles were identified from the five selected databases. After removing duplicates, 314 studies were retained. Upon reviewing the abstracts, 169 articles were excluded due to insufficient data and/or differing study designs. Consequently, 145 full-text articles were further assessed, and 72 articles were ultimately selected for inclusion. The PRISMA flowchart outlining the article selection process for this systematic review is presented in Figure 1.

## 3. Results and Discussion

Key points of correlation between AMH levels and metformin hydrochloride therapy in women with PCOS and IR are shown in Table 1. Numerous metabolic and endocrine disorders are observed in patients with polycystic ovary syndrome, which actively contribute to the syndrome’s pathophysiology. Studies have demonstrated that women with PCOS exhibit significantly elevated serum levels of anti-Müllerian hormone compared to healthy controls [28,38,39]. A positive correlation between serum AMH levels and androgen concentrations, specific to PCOS, has also been identified [40]. In this context, AMH contributes to the development and maintenance of intrafollicular hyperandrogenism by reducing aromatase activity and decreasing granulosa cell sensitivity to follicle-stimulating hormone [41].

Insulin plays a significant role in the pathophysiology of polycystic ovary syndrome by influencing androgen production through several mechanisms:

Stimulation of Ovarian Androgen Production: Hyperinsulinemia, often resulting from insulin resistance, common in PCOS, can directly stimulate ovarian theca cells to produce androgens. This effect is compounded by insulin’s ability to enhance luteinizing hormone (LH) activity, further promoting androgen synthesis [42].

Stimulation of Adrenal Androgen Production: Insulin resistance and the accompanying hyperinsulinemia may also stimulate the adrenal glands to secrete excess androgens. Studies have shown a correlation between insulin sensitivity indices and adrenal androgen responses, suggesting insulin’s role in adrenal hyperandrogenism [43].

Reduction in Sex Hormone-Binding Globulin (SHBG): Elevated insulin levels suppress hepatic production of SHBG, a protein that binds to androgens in the bloodstream, rendering them inactive. Lower SHBG levels result in increased free (active) androgens, exacerbating hyperandrogenic symptoms in PCOS [44].

These mechanisms contribute to the elevated androgen levels observed in individuals with PCOS, leading to clinical manifestations such as hirsutism, acne, and menstrual irregularities.

The active component of these androgens, free testosterone, rises when insulin directly reduces serum levels of sex hormone-binding globulin [45]. Additionally, insulin enhances ovarian responsiveness to gonadotropins by increasing the sensitivity of pituitary gonadotropes to gonadotropin-releasing hormone (GnRH). These findings underscore the crucial role of insulin in PCOS and the clinical evidence of insulin resistance and hyperinsulinemia in PCOS patients. Consequently, it is worth investigating whether metformin-induced reductions in hyperandrogenism are associated with changes in serum AMH levels.

Women with PCOS and pronounced insulin resistance often exhibit higher AMH levels due to insulin’s stimulatory effect on ovarian granulosa cells, increasing AMH secretion. By reducing insulin resistance, metformin diminishes this effect. In patients with PCOS and significant insulin resistance, metformin therapy is more effective in lowering AMH levels compared to those without insulin resistance, highlighting the role of insulin sensitization in improving ovarian function. Insulin resistance plays a critical role in regulating AMH in women with PCOS, and metformin’s ability to improve insulin sensitivity leads to the normalization of metabolic and hormonal imbalances, including reduced AMH levels. Thus, AMH can serve not only as a marker of ovarian function but also as an indicator of metformin therapy effectiveness in women with PCOS and insulin resistance.

Insulin resistance significantly influences anti-Müllerian hormone levels in women with polycystic ovary syndrome. Elevated insulin levels can stimulate ovarian granulosa cells to produce more AMH, leading to higher serum AMH concentrations. Metformin, an insulin-sensitizing agent, improves insulin sensitivity and has been associated with reduced AMH levels in some studies. For instance, a meta-analysis reported that metformin treatment resulted in a significant decrease in AMH levels among PCOS patients, particularly in those under 28 years of age [24].

However, the impact of metformin on AMH levels may vary based on factors such as body weight. A study observed that metformin led to a significant decrease in AMH levels in obese and overweight women with PCOS, while no significant change was noted in normal-weight women [46].

Furthermore, insulin resistance appears to modulate AMH production. Research suggests that insulin directly influences the cytochrome P450c17 enzyme system, enhancing androgen production within granulosa cells [47]. Therefore, metformin’s role in improving insulin sensitivity may contribute to normalizing AMH levels and restoring ovarian function in women with PCOS and significant insulin resistance.

Insulin resistance plays a crucial role in regulating AMH levels in PCOS. Metformin’s ability to enhance insulin sensitivity can lead to a reduction in AMH levels, potentially serving as an indicator of treatment effectiveness.

Metformin treatment in PCOS patients has shown significant reductions in serum AMH levels, with prolonged therapy (four months) required to suppress AMH effectively. Several studies confirm that metformin, as an insulin sensitizer, decreases AMH levels [24,36]. AMH levels in women with PCOS can serve as a marker for assessing the effectiveness of metformin treatment.

Further research is needed to fully elucidate the role of AMH in the development of PCOS, potentially paving the way for improved clinical approaches to managing the syndrome.

AMH is an important factor for evaluating ovarian reserve and is often elevated in women with PCOS. Metformin hydrochloride is widely used in PCOS therapy, particularly in women with insulin resistance. The impact of metformin on AMH levels in PCOS is significant, as it leads to a substantial reduction in serum AMH levels, suggesting improvements in polycystic ovarian morphology [24].

Palomba et al. demonstrated that metformin treatment significantly reduced AMH levels in women with PCOS after 6 months of therapy, indicating that the drug could help restore more typical ovarian function and balance the follicular reserve [21].

Teede et al. also found that metformin had a beneficial effect on reproductive outcomes in women with PCOS by reducing ovarian hyperandrogenism and improving ovulation rates. This study suggests that AMH levels may decrease as part of the broader normalization of ovarian activity induced by metformin [22].

Liu et al. further confirmed that metformin reduced AMH levels by decreasing the number of small antral follicles, which are a key source of AMH production in women with PCOS [23].

The dosage and duration of treatment vary, with lower doses of metformin (≤1500 mg/day) and shorter treatment durations (≤12 weeks) being associated with more pronounced reductions in AMH levels [48].

Age also plays a role, as younger patients (under 28 years) show a greater decrease in AMH levels in response to metformin therapy [24]. Foroozanfard et al. found that metformin treatment significantly reduced AMH levels in women with PCOS after an eight-week period, suggesting improvements in follicular dynamics and ovulation [49].

Gihan et al. [50] examined the effect of metformin on AMH levels in women with PCOS and insulin resistance. The results indicated a significant decrease in AMH levels after three months of therapy, suggesting improved ovarian function [50]. Long-term use of metformin (≥6 months) shows sustained reductions in insulin resistance, androgen levels, and AMH.

Another study published in BMC Research Notes highlighted a positive correlation between AMH levels and homeostasis model assessment of insulin resistance (HOMA-IR) in women with PCOS, emphasizing AMH’s potential as a biomarker for assessing metabolic status in these patients [51].

As an insulin sensitizer, metformin improves insulin sensitivity, reduces hyperinsulinemia, and suppresses hyperandrogenism. This process inhibits the growth of antral follicles, reflected in decreased AMH levels. Studies indicate that metformin regulates insulin and androgen levels, leading to reduced ovarian hyperandrogenism and, ultimately, lower AMH levels [48]. Insulin resistance acts as a moderating factor, as PCOS patients with higher insulin resistance experience a more significant reduction in AMH following metformin therapy compared to those without insulin resistance [24].

Analysis of data in the literature reveals additional relationships between AMH, insulin resistance, and polycystic ovary syndrome, which we can highlight.

**Table 1 healthcare-13-00884-t001:** Key points of correlation between AMH levels and metformin hydrochloride therapy in women with PCOS and IR.

Key Benefit/Topic	Significance	Authors
1. Reduction in AMH levels	The reduction in AMH levels reflects an improvement in folliculogenesis and ovarian function, indicating a positive therapeutic response.	Palomba et al., 2011 [21]; Teede et al., 2010 [22]; Zhou et al., 2023 [24]; Liu et al., 2017 [23]; Pigny et al., 2006 [28]; Neagu & Cristescu 2012 [36]; Karimzadeh 2016 [46]; Di Lorenzo et al., 2023 [38]; La Marca et al., 2004 [39].
2. Improvement in insulin sensitivity	Metformin enhances insulin sensitivity by lowering blood glucose levels and decreasing hyperinsulinemia, which indirectly affects AMH levels.	Teede et al., 2010 [22]; Zhou et al., 2023 [24]; Mehdinezhad et al., 2024 [48]; Gihan et al., 2024 [50]; Wiweko et al., 2018 [51]; Rojas et al., 2014 [52]; Karkanaki et al., 2011 [53].
3. Correlation with androgen reduction	Elevated AMH levels in PCOS are often associated with hyperandrogenism. Metformin’s reduction of androgens correlates with a decrease in AMH, as reduced androgenic stimulation normalizes ovarian function.	Wiweko et al., 2014 [29]; Catteau-Jonard et al., 2007 [40]; Orio et al., 2005 [41]; Mansour et al., 2024 [42]; Stogowska et al., 2023 [43]; Xing Diamanti-Kandarakis et al., 2022 [44]; 2010; Bulun S.E. 2016 [45]; Su et al., 2025 [47]; Harada et al., 2022 [54].
4. Normalization of follicular dynamics	Metformin helps restore normal follicular dynamics by modulating pathways involved in cell proliferation, such as the mTOR pathway. This can reduce the excessive follicular recruitment seen in PCOS, leading to a decrease in AMH levels.	Liu et al., 2017 [23]; Zhou Z. et al., 2023 [24]; Orio et al., 2005 [41]; Rojas et al., 2014 [52]; Ding et al., 2021 [55]; Usman et al., 2020 [56].
5. Predictive value for treatment success	Monitoring AMH levels during metformin therapy serves as a predictive biomarker of treatment success, allowing clinicians to assess whether the therapeutic strategy is effective. A consistent decline in AMH during therapy indicates improved ovarian and metabolic function, while persistently high levels may suggest the need for additional therapeutic approaches.	Mehdinezhad et al., 2024 [48]; Nelson et al., 2023 [57]; Stańczak et al., 2024 [25]; Neve-Dolfing 2012 [58]; 72. Malhotra et al., [59].
6. Individual variability and personalized treatment	The degree of AMH reduction varies among women, depending on baseline insulin resistance, severity of PCOS, and individual responsiveness to metformin. Personalizing metformin therapy by regularly measuring AMH levels ensures more targeted and effective management of PCOS and IR.	Mehdinezhad et al., 2024 [48]; Legro RS., 2000 [60]; Sánchez-Garrido et al., 2024 [61]; Tan et al., 2007 [62].

### 3.1. AMH as a Biomarker of the Severity of PCOS

Women with PCOS typically exhibit elevated anti-Müllerian hormone (AMH) levels, which correlate with the number of antral follicles, reflecting disrupted follicular growth and increased ovarian activity characteristic of PCOS.

Insulin resistance leads to hyperinsulinemia, which stimulates ovarian granulosa cells, thereby increasing AMH production. Higher AMH levels are often associated with more severe IR in women with PCOS [10,52,53] (Table 2).

### 3.2. AMH and Metabolic Risk in PCOS

Elevated AMH levels are linked to an increased risk of metabolic syndrome in PCOS, including dyslipidemia, hypertension, and abdominal obesity. AMH levels significantly decrease following metformin therapy, particularly in obese PCOS patients, while reductions in non-obese patients are less pronounced. There is a significant relationship between AMH levels and the antral follicle count, with higher AMH levels being associated with a 1.5-fold increase in follicle numbers [52,55,56]. 

IR contributes to the development of metabolic syndrome, and AMH levels may serve as an indicator of metabolic risk in women with PCOS.

### 3.3. Hormonal Influence on AMH and IR

Hyperandrogenism represents one of the potential hormonal effects influencing AMH. Elevated androgen levels in PCOS stimulate ovarian granulosa cells, leading to increased AMH secretion. IR exacerbates hyperandrogenism through hyperinsulinemia, creating a vicious cycle. 

AMH regulates luteinizing hormone secretion by affecting the hypothalamic–pituitary axis. This can exacerbate the imbalance between LH and FSH commonly observed in PCOS [29,54].

### 3.4. Impact of Treatment on AMH and IR

As previously noted, metformin improves IR, leading to a reduction in AMH levels. This is particularly beneficial for women with PCOS aiming to reduce follicular hyperactivity and restore ovulation.

Combining metformin with other therapies targeting IR may further decrease AMH levels, improving hormonal balance and ovarian function [25,58] (Table 2).

### 3.5. AMH and Fertility in PCOS

High AMH levels are often associated with anovulation, a hallmark of PCOS. Treating IR with metformin can reduce AMH levels and improve ovulation.

As a prognostic factor, AMH reduction following therapy indicates successful ovulation induction and improved fertility. For a characterization of women with elevated anti-Müllerian hormone levels (AMH) and the correlation of AMH with polycystic ovarian syndrome phenotypes and assisted reproductive technology outcomes, see Table 2 [1,6,63].

### 3.6. Long-Term Consequences of IR, AMH, and PCOS

IR in PCOS increases the risk of developing type 2 diabetes. Women with PCOS, elevated AMH levels, and IR are at higher risk of cardiovascular diseases. 

The interplay between AMH, IR, and PCOS reflects both hormonal and metabolic dysfunctions in the body. AMH may serve as an indicator of PCOS severity, treatment efficacy, and the risk of long-term complications. Therapies improving IR play a crucial role in normalizing these biomarkers [47,52,64].

### 3.7. Monitoring IR and PCOS in Metformin Therapy

#### 3.7.1. Central Mechanism in PCOS Pathophysiology

IR is a key factor in PCOS development, with hyperinsulinemia stimulating ovarian theca cells, leading to hyperandrogenism and hormonal imbalances that exacerbate PCOS symptoms, including irregular cycles, anovulation, and hirsutism [65].

Metformin reduces IR, suppresses hyperinsulinemia and hyperandrogenism, and addresses the root causes of PCOS rather than merely alleviating symptoms [48,66,67] (Table 2).

#### 3.7.2. Therapeutic Efficacy

Optimal metformin therapy improves both metabolic and hormonal imbalances in women with PCOS and IR. Monitoring the relationship between IR and PCOS helps assess whether metformin is the most appropriate treatment for an individual patient. 

Women with pronounced IR demonstrate better responses to metformin, while those with minimal IR may require additional or alternative therapies (e.g., myo-inositol and combination treatments) [13,61,62] (Table 2).

#### 3.7.3. Improvement in Reproductive Function

IR is a major contributor to anovulation in PCOS. By regulating insulin and androgens, metformin enhances ovulation.

Decreased AMH levels following metformin therapy are associated with the restoration of normal follicular dynamics and increased chances of conception [48,55,66].

#### 3.7.4. Prevention of Long-Term Complications

IR significantly increases the risk of type 2 diabetes in women with PCOS. Metformin therapy mitigates this risk by improving insulin sensitivity. 

Monitoring IR provides insight into metformin’s effects on cardiometabolic risks associated with PCOS, such as cardiovascular disease and metabolic syndrome [68].

#### 3.7.5. Biomarker-Based Monitoring

Indicators such as AMH and insulin dynamics are critical for assessing therapeutic efficacy. Reductions in AMH levels and improvements in IR markers (e.g., HOMA-IR) signal successful treatment with metformin. 

Continuous monitoring of IR, AMH, and other biomarkers (e.g., testosterone, glucose, and insulin) allows clinicians to track progress and adjust therapies accordingly [69].

#### 3.7.6. Enhanced Quality of Life

Addressing IR alleviates PCOS symptoms, such as hirsutism, acne, and excess weight, significantly improving self-esteem and overall quality of life. 

Early detection of the IR-PCOS connection helps prevent chronic complications and improves overall health outcomes.

PCOS is a multifaceted condition with lifelong implications and is increasingly prevalent among women of reproductive age. The primary challenges in managing this syndrome lie in its ambiguous diagnostic criteria and the complexity of its symptoms. Adopting timely, personalized treatment strategies significantly improves overall management, reduces comorbidities, and enhances quality of life. Early detection and intervention, particularly for women at risk of infertility, are crucial for achieving better outcomes and prognoses [6,70,71].

## 4. Conclusions

These studies highlight the importance of monitoring insulin resistance (IR) and AMH levels in women with PCOS undergoing metformin therapy. Reductions in AMH levels and improvements in insulin sensitivity can serve as indicators of treatment efficacy and enhancements in reproductive function for these patients.

AMH could be considered a prognostic marker for evaluating the effectiveness of metformin therapy. A decrease in AMH levels following treatment may indicate improved ovarian function and a reduction in polycystic morphology. However, further research is necessary to confirm these findings and to determine the optimal dosages and duration of treatment.

## Figures and Tables

**Figure 1 healthcare-13-00884-f001:**
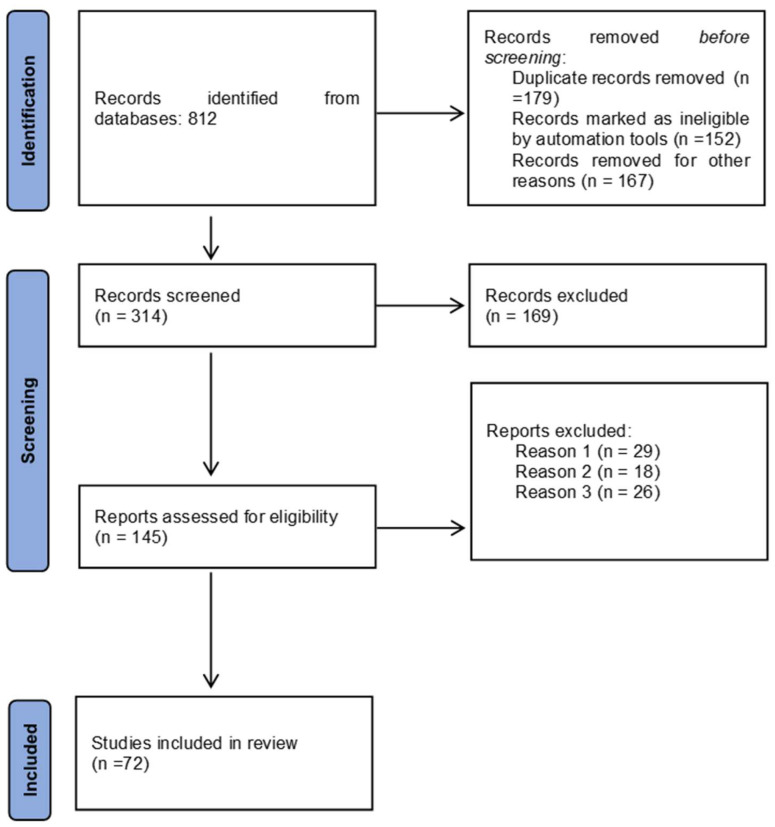
PRISMA flowchart.

**Table 2 healthcare-13-00884-t002:** Key benefits of AMH as a biomarker of metformin treatment effectiveness and personalized therapeutic strategies in PCOS and IR management.

Key Benefit/Topic	Significance	Authors
1. Assessment of ovarian reserve and follicular status	Metformin’s ability to reduce AMH levels may indicate a decrease in the excessive number of small follicles, reflecting improved ovarian function.	Palomba et al., 2011 [21]; Liu et al., 2017 [23]; Zhou et al., 2023 [24]; Orio et al., 2005 [41]; Rojas et al., 2014 [52]; Ding et al., 2021 [55]; Usman et al., 2020 [56].
2. Monitoring therapeutic response	This biomarker allows clinicians to make data-driven decisions about continuing or adjusting the therapeutic regimen.	Teede et al., 2010 [22]; Mehdinezhad et al., 2024 [48]; Legro RS., 2000 [60]; Sán-chez-Garrido et al., 2024 [61]; Tan et al., 2007 [62].
3. Personalizing treatment strategies	Personalized therapeutic approaches based on AMH dynamics during treatment can optimize outcomes, ensuring that patients receive the most suitable intervention.	La Marca et al., 2010 [39]; Mehdinezhad et al., 2024 [48]; Legro RS., 2000 [60]; Sán-chez-Garrido et al., 2024 [61]; Tan et al., 2007 [62].
4. Predicting ovulation and fertility outcomes	Since metformin improves ovulation rates in women with PCOS, decreasing AMH levels can indicate enhanced ovulatory function. This can be particularly useful when planning fertility treatments or predicting spontaneous ovulation.	Liu et al., 2017 [23]; Zhou et al., 2023 [24]; Witchel et al., 2019 [1]; Singh et al., 2023 [6]; Tal et al., 2014 [63]; Orio et al., 2005 [41].
5. Assessing metabolic improvement	Using AMH as a biomarker in conjunction with metabolic parameters (like fasting insulin and glucose levels) provides a comprehensive picture of therapeutic success.	Rojas et al., 2014 [52]; Ding et al., 2021 [55]; Usman et al., 2020 [56]; Nelson et al., 2023 [57].

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
