# Peer review of "The Anti-Mullerian Hormone as a Biomarker of Effectiveness of Metformin Hydrochloride Therapy in Polycystic Ovarian Syndrome and Insulin Resistance"

_healthcare, 2025, doi:10.3390/healthcare13080884_

Round 1
Reviewer 1 Report
Comments and Suggestions for Authors
In this review, the authors investigated the possibility of AMH being a biomarker for predicting the therapeutic effects of metformin therapy in patients with PCOS accompanied by IR through a systematic review, and concluded that AMH is in fact a prognostic marker. They also concluded that further research is needed to determine the optimal dosage and duration of administration.
AMH measurement is widely used as a standard test in the diagnosis of PCOS and in the therapeutic management of metformin therapy for PCOS. In this situation, the rationale for conducting a systematic review in this review is lacking. Clarifying this rationale would greatly enhance the value of this review.
In addition, the abstract and conclusion state that further research is needed to determine the optimal dosage and duration of administration, but the reasons for this need are not clearly stated.
Furthermore, the structure of the description needs to be greatly improved. The logic is very difficult to follow.
In addition, there are points that need improvement and points that are unclear, as listed below.
#Would it be possible to briefly show the diagnostic criteria for PCOS (Line 42)?
#The description of the function of AMH is only shown in Lines 50-52. Shouldn't you introduce this by citing a few more references?
# The statement ‘AMH decreases with metformin treatment’ is made in Lines 168, 177, 182, and 202, citing Zou et al 2023, but in Line 62-64, it is described by citing the paper by Nascimento et al. In the sentence immediately before that, it is stated that ‘its effect on AMH levels and ovarian function remains controversial’ by citing Zhou et al. 2023. There is no consistency in the way citations are made or in the way they are described. Why is this?
#It is stated in Lines 65 and 67-69 that ‘the concentration of serum AMH levels are higher in women with PCOS’, but there is a similar statement in Lines 121-130. However, the references cited are different. Why is this?
#The abbreviation for polycystic ovary syndrome is already stated as PCOS in Line 41, but it is also indicated many times after that as Polycystic ovary syndrome (PCOS). In some cases, it is only PCOS. This should be unified. The same applies to IR and SHBG.
#The sentence in lines 103-105 (By examining changes in AMH as a response to Metformin therapy, the research seeks to elucidate potential mechanisms by which Metformin improves ovarian function and overall reproductive health in women with PCOS. ) is also described as the purpose of this research, but is this point ultimately discussed?
#In line 135, it is stated that ‘there is a strong, positive correlation between LH and AMH levels’ (what is the reference?), but in line 136, there is a contradictory statement that ‘women with PCOS exhibit insufficient LH’. I don't really understand the reason for this.
#What is the reference for lines 137-140?
#There is a statement that androgen production increases when adrenal androgen synthesis is suppressed (Line 144-146), but it is difficult to understand the reason for this. I would like a little more explanation.
#Is the statement in Line 149-150 based on the research of Nestler et al 1996?
#Line 152-154 is abrupt when considering the previous statement.
#Line 155-164: Suitable references should be cited.
# Are there any references related to Lines 165-167?
#Is ‘elevated’ in Line 168 necessary?
#Are the sentences in Line 170-172 necessary here?
#What is the percentage shown in Figure 1 a percentage of?
#From line 206 onwards, there are hardly any references cited. It is difficult to tell which sentences are based on previously presented findings and which the author wants to emphasize. Appropriate references should be cited for appropriate sentences.
#Line 250-291 3.5 --> 3.7
#Line 250-291: There is almost no description of AMH. Should these descriptions clearly show what the relationship is with AMH?
Author Response
Response to REVIEWER 1
- We included diagnostic criteria for PCOS (Line 42).
- Lines 50-52. We cited a few more references.
- The citation by Nascimento et al is changed; other citation are put in order (line168 – 202); citations of Zhou et al. are redused
- Lines 65 and 67-69 – citations are for the introduction and Lines 121-130 are for the purpose of discussion section. Now it is transformed because of changing the review by PRISMA systematic method.
- All abbreviations are unified - PCOS, AMH ets.
- Lines 135 – 140 are removed.
- Line 144-146 – we made new explanation.
- Line 149-150 – we changed citation.
- 156 – 165 are connected to citations.
- Line 168 - ‘elevated’ is removed.
- Figure 1 is removed. There is new figure 1 PRISMA flowchart.
- Line 250-291 3.5 --> 3.7 – numbers are corrected.

Reviewer 2 Report
Comments and Suggestions for Authors
Thank you very much for inviting me to contribute to the review of this very interesting literature review. The idea behind the work is very interesting. Metformin treatment for women with policystic ovaries is now very readily available and widely used. Therefore, the insights regarding the correlation of metformin treatment with ovarian reserve are very important. Higher doses of metformin are often used in the treatment of PCOS, compared to the treatment of normal insulin resistance and other glycemic disorders.
The manuscript generally holds up well but there are a few issues that would require attention and possible correction:
1. First of all, the methodology section lacks a thorough description of the literature collection. I suggest that the section be expanded to describe the exact method of literature review, for example, using the PRISMA scheme. Describe what databases were searched and how. How many publications were used to conduct the review, etc....
2. There is a lack of discussion of the form (long-acting-short-acting) and dose of insulin used in PCOS as well as the time of treatment.
3. What I do not understand is the graph shown in Fig. 1. On the basis of which studies these figures were established. Where did they come from. If the graph was taken from another publication, the source should be given and permission obtained to present it. If the graph is your own, you should absolutely show where it came from and how the relevant percentages were calculated.
4. Consideration could be given to adding appropriate tables, where the individual publications included in the current review would be listed, where, for example, the number of participants in the study would be given, as well as the main results of the study in question, and perhaps other data.
Author Response
Response to REVIEWER 2
- We prepared this review according to the requirements of PRISMA systematic model.
- Insulin administration is not considered in this review, we consider metformin therapy administration.
- We removed figure 1. There is a new figure 1 PRISMA flowchart.
- We included a tables with the key benefits/topics, according to the requirements of PRISMA model.

Author Response
Response to REVIEWER 3
This review is not relevant to our article!

Reviewer 4 Report
Comments and Suggestions for Authors
This work by Nikoleta Parahuleva et al, is a review that highlights the anti-mullerian (AMH) hormone potential as an improvement biomarker of PCOS and IR under metformin treatment. The authors made a good literature analysis with a manuscript that is easy to read and understand. This reviewer has the next suggestions to enhance some points:
Line 51. It is suggested to specify more the central role of AMH, how it works under normal conditions, how it is regulated, etc.
Line 92,99,101. It is suggested to change the word "verbal" to oral, which is more formal when referring to the route of drug administration.
Line 92. It is suggested to verify if the word “sedate” is correct.
Line 99. Define MIS before its abbreviation.
Methodology: Even though this review responds to a research question the methodology should be more specific since this work is a systematic review, it is suggested that authors describe how many papers were found and how many were evaluated and considered, which inclusion and exclusion criteria were established, the authors could show this information in a diagram form.
Also, in methodology if the authors employed a quality evaluation or a guide to realize this systematic review should be declared.
This reviewer suggests to the authors consider the PRISMA statements for reporting systematic reviews: BMJ 2021;372:n71 http://dx.doi.org/10.1136/bmj.n71
Results: Is Figure 1 about a particular study referred to? It is suggested to mention this figure in a paragraph with more detail. In case Figure 1 summarizes the results of several studies, a meta-analysis could be performed and a forest plot graph could be presented for each metformin effect showed
It is also suggested to summarize in a table the different articles considered, to identify the sample size of each study, the dose and time of metformin, HOMA values, insulin, hormones, age of participants, and main results.
Comments on the Quality of English LanguageRefer to metformin as a drug or antidiabetic oral instead of "verbal" is more formal
Author Response
Response to REVIEWER 4
Line 51. We included a paragraph that specify more the role of AMH, how it works under normal conditions, and how it is regulated
Line 92,99,101. We changed the word "verbal" to oral or cut it.
Line 99. Abbreviation is changed to proper US FDA - Food and drug agency
We removed figure 1. There is a new figure 1 PRISMA flowchart.
We prepared this review according to the requirements of PRISMA systematic model.
We included a tables with the key benefits/topics, according to the requirements of PRISMA model.

Round 2
Reviewer 1 Report
Comments and Suggestions for Authors
The authors did not respond appropriately to some of my comments made in the previous review.
Tables 1 and 2 have been added. These are thought to be important points of this review. However, there is no mention of these tables in the main text. The results and discussion should be developed using these tables.
The final chapter, 3.7, should be an important chapter that concludes this paper, but it is difficult to derive a conclusion (4. Conclusion) from the explanation.
In general, the development of the logic is very difficult to understand.
Also, in Tables 1 and 2, the numbers of the cited references should be clearly stated together with the names of authors.
Author Response
We linked tables 1 and 2 to the text. We also expanded the related references so that the results and discussion are clearer, we think this has greatly improved the connection and logic of the presented results, for which we thank you!

Reviewer 2 Report
Comments and Suggestions for Authors
Thank you for inviting me to review this article.
In the new version of the manuscript, the authors have met all my expectations.They have put a lot of effort to improve and enhance the initial version. I have no further comments. I am in favor of accepting the text in its current form for publication.
Author Response
Thank you!
Reviewer 4 Report
Comments and Suggestions for Authors
The authors have improved the manuscript, and the dedicated work is evident. However, there are some points related to the methodology and results that need to be improved:
- Literature search: why did the authors consider as relevant studies of molecular classification of endometrial carcinoma and its precursor lesions instead of PCOS and IR, and metformin? justify
- Eligibility Criteria: The eligibility criteria only consider the year and type of publication. In the case of original articles, did the authors consider metformin dose/time? age of participants? preclinical and clinical studies? It is important to specify this information.
- PRISMA flowchart: Is it suggested to describe the reasons 1, 2 and 3 of the excluded reports.
- Table 1 and 2: Although tables 1 and 2 summarize findings related to AMH, metformin, IR, and PCOS, it is suggested to describe in another(s) column(s) the type of publication and study design from which the information comes, metformin dose, time, age of the participants, sample size.
- The methodology states that 72 articles were evaluated for this review, the majority should be cited between tables 1 and 2, however, few articles are cited in tables, and even the bibliography in this article only includes 71 references. It is suggested to add a table with the information on each article evaluated and its reference.
- In the references included in tables, in addition to the author's name, indicate the reference number since the bibliography is not written in alphabetical order.
Author Response
The study of the molecular classification of endometrial carcinoma and its precursor lesions is the subject of another study of ours.
The reasons for excluded reports are usually: only a summary, lack of full text, and inability to translate or adapt.
We linked tables 1 and 2 to the text. We also expanded the related references and linked them to the reference number, so that the results and discussion are clearer, we think this has greatly improved the connection and logic of the presented results, for which we thank you!
